# The Weekend Effect on In-Hospital Mortality—First 13-Year Retrospective Observational Study in Slovakia

**DOI:** 10.3390/healthcare13121412

**Published:** 2025-06-12

**Authors:** Orsolya Hrubá, Lucia Žigová, Michala Hrončová, Simona Valášková, Juraj Smaha, Peter Jackuliak, Martin Kužma, Alexander Mayer, Andrej Dukat, Juraj Payer, Jan Kyselovic, Andrea Gažová

**Affiliations:** 1Institute of Pharmacology and Clinical Pharmacology, Faculty of Medicine, Comenius University Bratislava, 81372 Bratislava, Slovakia; orsi.hruba@gmail.com (O.H.); lucia.zigova38@gmail.com (L.Ž.); michala.smelkova@gmail.com (M.H.); aandreagazova@gmail.com (A.G.); 2International Laser Centre, Slovak Centre of Scientific and Technical Information, 84104 Bratislava, Slovakia; 35th Department of Internal Medicine, Faculty of Medicine, Comenius University Bratislava, 81372 Bratislava, Slovakia; jurajsmaha@gmail.com (J.S.); peter.jackuliak@gmail.com (P.J.); kuzma.martin1@gmail.com (M.K.); andrej.dukat@fmed.uniba.sk (A.D.); prof.payer@gmail.com (J.P.); kyselovic@gmail.com (J.K.); 4Department of Surgery, Faculty of Medicine, Comenius University Bratislava, 81372 Bratislava, Slovakia; alexandermayer.ba@gmail.com; 5Department of Pharmacology and Toxicology, University of Veterinary Medicine and Pharmacy, 04181 Košice, Slovakia

**Keywords:** weekend admissions, weekend effect, mortality rate, retrospective cohort

## Abstract

Objectives: The “weekend effect” refers to the higher risk of death for patients admitted to hospitals on weekends compared to weekdays. While this pattern is well documented in many countries, there is limited data from Central Europe, including Slovakia. Study Design: We conducted a 13-year retrospective study at a large internal medicine department in Bratislava, Slovakia, to determine (1) whether there is a weekend effect, (2) which patient groups are most affected, and (3) what clinical and demographic factors might play a role. Methods: Using statistical tests, we analysed 45,955 hospitalisations between 2010 and 2022, comparing mortality rates between weekday (Monday–Friday) and weekend (Saturday–Sunday) admissions. We also used logistic regression to adjust for age, gender, length of stay, and re-hospitalisation. Causes of death were classified using ICD-10 codes. Results: Patients admitted on weekends had a significantly higher in-hospital mortality rate (15.58%, 1203 deaths among 7719 admissions) than patients admitted on weekdays (10.47%, 4002 deaths among 38,236 admissions, *p* < 0.0001). Even after adjusting for other factors, weekend admission remained a strong predictor of death (adjusted odds ratio = 1.31, 95% CI: 1.22–1.41). Cardiovascular disease was the most common cause of death on weekends. The weekend effect persisted across all years and was particularly pronounced in the COVID-19 pandemic (2020–2021). Conclusions: Our findings confirm the weekend effect in Slovakia, with patients admitted on weekends facing a higher risk of in-hospital death. Identifying the reasons behind this trend is critical to improving hospital care and ensuring consistent quality of care throughout the week.

## 1. Introduction

The “weekend effect” is a well-known phenomenon in healthcare, characterised by a higher mortality rate among those admitted to hospital at weekends than on weekdays. This phenomenon raises concerns about possible differences in healthcare quality and availability of resources depending on the day of admission.

The concept was first identified in a landmark 2001 study that found significantly higher hospital mortality among patients admitted on weekends [1]. While this effect has been extensively studied in the healthcare systems of North America [2,3,4], Western Europe [5,6,7,8,9,10,11,12], and Australia [13,14], little data is available from Central and Eastern Europe. However, studies have been conducted in regions such as Italy [15], Ireland [16], Spain [17,18], France [19], the Netherlands [20], and the Nordic countries [21,22,23,24,25,26]. Only a few studies are available from the Central European countries [27,28,29,30,31,32]. Studies from Austria and Poland reported significantly higher mortality for weekend admissions.

In contrast, single-diagnosis studies from Romania and Hungary have reported no significant difference in patient outcomes between weekends and weekdays. These contradictory results emphasise the need for more detailed, context-specific research in Central and Eastern Europe—regions that remain underrepresented in the literature on weekend effects. For Slovakia in particular, there are currently no such findings. To confirm this gap, we conducted a systematic search in PubMed, Scopus, and Web of Science using the terms “weekend effect” OR “weekend admission” AND “mortality” AND “Slovakia” (last search on 4 May 2025). The search returned zero articles examining in-hospital mortality by day of admission in Slovakia.

This lack of data highlights the urgency of regional studies that can shed light on how local hospital processes, patient distribution patterns, and severity of illness influence weekend mortality rates.

Extensive research has now confirmed that this effect persists in different healthcare systems. A comprehensive meta-analysis from 2019 covering over 640 million admissions concluded that the mortality rate increased by 16 (OR = 1.16) for weekend admissions compared to weekdays. A cross-national analysis also found that the variation in hospital mortality between weekdays is a systematic global phenomenon, although administrative data alone cannot reveal the causes [33].

From the patient’s perspective, the day of admission should not influence their chances of survival. However, the weekend effect suggests that systemic factors—such as staffing, diagnostics, or delays in decision-making—may affect care during specific periods. This raises concerns about the equity and consistency of healthcare. While the weekend effect is well-documented in Western healthcare systems, it has not yet been thoroughly investigated in Central and Eastern Europe. In Slovakia, a country with almost universal healthcare coverage, persistent problems such as labour shortages, regional inequalities, and underfunding can affect the quality and outcomes of care [34]. These structural issues can affect the quality and consistency of hospital care at weekends, emphasising the importance of studying the effect of the weekend in this national context.

Our study has three main objectives: (1) to determine whether there is a weekend effect in Slovakia, (2) to identify the patient groups most affected by higher weekend mortality, and (3) to investigate clinical or demographic factors associated with higher weekend mortality.

## 2. Materials and Methods

### 2.1. Study Setting and Data Sources

For this retrospective observational study, the data of all patients admitted to the 5th Department of Internal Medicine in Bratislava, Slovakia, between 2010 and 2022 were used. It is the largest internal medicine clinic in the city, providing comprehensive diagnostic and therapeutic care for a wide range of internal diseases. It has 86 beds, including an intensive care unit with 8 beds. The department admits acute patients from the Bratislava region but also offers elective admissions for patients from all over Slovakia [35]. The unidentified raw data at the patient level was taken from the hospital’s internal database, which routinely records hospitalisations, re-hospitalisations, and deaths in the hospital as part of quality monitoring. This database was implemented in 2010 following the introduction of the e-health system at the university hospital.

The analysis focused on hospital admissions where the patient’s length of stay was 30 days or less and where the patient died or was discharged within this period. We included admitted patients regardless of age, type of admission (elective or emergency), medical speciality, or diagnosis upon admission. We excluded admissions with more than 30 days of hospitalisation from the original dataset. We retrieved electronic medical records and integrated them with the administrative database to augment the dataset using RStudio (version 4.2.2, The R Foundation, Vienna, Austria) and Python (Version 3.10.6, Python Software Foundation, Wilmington, DE, USA). This integration process allowed us to include relevant patient-level clinical information, including demographics, primary and secondary diagnoses, admission details, lab values, and cause of death. To ensure the integrity of the data, we identified and removed all duplicate records.

The final structured dataset was then compiled in Microsoft Excel and prepared for statistical analysis and visualisation. This comprehensive dataset included various patient-level variables, as follows:
-Demographic data: age, gender, and two age groups (under 65 and over 65).-Admission characteristics: day of admission (Monday–Sunday), length of hospital stay (LOS), and re-hospitalisations within 30 days.-Clinical information: primary diagnosis, principal diagnosis, admission diagnosis, and cause of death—all coded according to the International Classification of Diseases, 10th Revision (ICD-10)-Results: in-hospital mortality within 30 days of hospitalisation.

In-hospital mortality was defined as death during the index hospitalisation at Bratislava University Hospital. Deaths occurring after discharge were not included in the dataset. As a result, this definition may lead to a bias related to length of stay, as longer hospitalisations increase the likelihood of a death being recorded. To mitigate this, length of stay was included as an adjustment variable in the logistic regression model.

### 2.2. Study Objectives

The primary aim of this study was to assess the presence and impact of the weekend effect on the internal medicine ward. We compared in-hospital mortality rates between weekday (Monday–Friday) and weekend (Saturday–Sunday) admissions.

We also wanted to understand how patient characteristics (age and gender) and clinical factors (diagnosis and length of stay) influence this outcome. We focused on identifying potential risk factors that contribute to the weekend’s impact.

### 2.3. Data Extraction

Two datasets were manually extracted from the hospital information of the University Hospital Bratislava and saved in Microsoft Excel format:(1)A comprehensive dataset of all hospitalisations from 2010 to 2022 in the 5th Department of Internal Medicine;(2)A detailed set of de-identified medical records of deceased patients.

The general dataset on hospitalisations contained information on admission and discharge dates, age, gender, length of stay, and outcome (discharge or death).

The medical records of deceased patients were more detailed and contained:-ICD-10 codes for admission and cause of death;-Complete summaries of medical care;-Clinical and functional history;-Diagnostic investigations, including laboratory results (haematology, biochemistry, microbiology, urinalysis);-Imaging examinations (X-ray, CT, ultrasound);-Comorbidities and chronic diseases;-Medication history and treatment measures during hospitalization;-Specialist consultations (e.g., surgery, urology, rehabilitation);-Final clinical course and summary of the dying process.

The two datasets were merged using R (version 4.2.2) using unique patient identifiers to ensure accurate linkage. A manual check and validation were performed to ensure consistency and eliminate duplicates. This integrated dataset was then used for descriptive and multivariable statistical analysis.

### 2.4. Statistical Analysis

We processed and statistically analysed the data using R (version 4.2.2, The R Foundation, Vienna, Austria) and Python (Version 3.10.6, Python Software Foundation, Wilmington, DE, USA). For the analysis in R, we used the following packages: ggpubr, tidyverse, ggthemes, compareGroups, FSA, car, readxl, psych, nplr, data.table, ggrepel, openxlsx, directlabels, digest, writexl, and xlsx. In Python, we used the following libraries: os, regex, pandas openpyxl, numpy, itertools, pathlib, unicodedata, datetime, operator, csv, nltk unidecode, and collection. We also used Microsoft Excel (Microsoft Corporation, 2021) to visualise the data. In our analysis, we applied a 95% confidence interval (CI) and set the threshold for statistical significance at *p* < 0.05.

We used descriptive statistics to summarise the patients’ baseline data. Continuous variables, such as age and length of stay, were expressed as means with standard deviations (SD). Categorical variables, including gender, day of admission, and cause of death, were presented as absolute frequencies and percentages. We analysed differences in in-hospital mortality rates between weekday and weekend admissions using a chi-square test to determine statistical significance.

We used a multivariable logistic regression model to determine whether weekend admission was an independent predictor of in-hospital mortality. This allowed us to estimate adjusted odds ratios (AORs) with 95% confidence intervals. The model was adjusted for key confounders, including age, gender, length of hospital stay, and re-hospitalisation status, with weekday admissions as the reference category.

We also conducted a cause-specific mortality analysis to determine whether certain conditions were disproportionately associated with weekend admissions. Causes of death were classified using ICD-10 coding [36], with a focus on cardiovascular disease, respiratory disease, sepsis, and malignancy. We applied a Haldane–Anscombe correction to avoid bias in cases where the number of events was low.

Finally, we performed a subgroup analysis to determine whether the weekend effect varied by age or gender. Interaction terms (weekend × age; weekend × gender) were included in the regression model. Neither interaction term was statistically significant, indicating that the weekend effect remained consistent across all demographic subgroups.

### 2.5. Ethical Considerations

The ethical principles of the Declaration of Helsinki guided this study. The analysis was based on fully anonymised, retrospective data from hospital records. According to the Statute of the Ethics Committee of the University Hospital Bratislava (Po-138/2015) [37], all patient records were de-identified before analysis to ensure privacy and data protection.

## 3. Results

Our data collection included administrative data of all patients admitted to the internal medicine department between 2010 and 2022, resulting in 46,521 cases. Within this dataset, 17,501 were identified as repeat hospitalisations, of which 3756 occurred within 30 days of initial discharge.

### 3.1. Patient Characteristics and Admission Patterns

We initially identified 46,521 hospital admissions at the 5th Department of Internal Medicine between 2010 and 2022. The final cohort, after excluding patients regarding the exclusion criteria, resulted in a final analytic sample of 45,955 patients (Figure 1), with a mean age of 69.0 ± 16.2 (72.1 [60.2–81.2]) years and an average length of stay (LOS) of 6.1 ± 5.2 days (5 [2–8]).

Since our database only contained the admission date of each patient without the exact time (based on date only), we defined the following operationally:-Weekday admissions as those occurring from Monday to Friday-Weekend admissions as those occurring from Saturday to Sunday.

Admissions were more frequent on weekdays than on weekends, with the lowest number of hospitalisations recorded on Saturdays and Sundays, based on these definitions. In terms of gender distribution, males accounted for 48.5% (22,288) of total admissions with a mean age of 66.1 ± 15.9 years (66.1 [56.8–78.4]), while females accounted for 51.5% (23,667) of admissions and had a higher mean age of 71.6 ± 16.0 years (70.4 [63.4–81.9]).

We compared baseline data for patients admitted on weekdays and weekends to exclude potential confounders. On weekends, patients admitted were slightly older than those admitted on weekdays (mean age 69.96 vs. 68.77 years; *p* < 0.0001), which is a statistically significant difference. However, the gender distribution between weekend and weekday admissions showed no significant difference (*p* = 0.139), suggesting that gender probably did not influence the observed weekend effect (Table 1).

We also analysed the distribution of diagnoses among the deaths by day of admission. The most common causes of death were related to cardiovascular and respiratory diseases, and this pattern was the same in both groups. The chi-square test comparing the overall distribution of diagnosis codes between weekday and weekend deaths showed no statistically significant difference (*p* = 0.615), suggesting that the case mix was similar regardless of the day of admission.

A total of 5205 in-hospital deaths occurred within 30 days, of which 2491 (47.9%) were men and 2714 (52.1%) were women. Most deaths occurred on Fridays, with 810 deaths. The overall mean age at death was 77.0 ± 11.8 years (79.2 [70.1–85.8]); males had a mean age of 74.6 ± 12.0 years (76.0 [67.1–83.5]), while females had a higher mean age of 79.2 ± 11.2 years (81.6 [73.2–87.1]) (Table 2).

### 3.2. In-Hospital Mortality

During the study period, a total of 5205 in-hospital deaths occurred within 30 days: 4002 deaths among 38,236 weekday admissions and 1203 deaths among 7719 weekend admissions. The average age at the time of death was 77.0 ± 11.8 years, and the average length of stay (LOS) was 6.5 ± 6.6 days. Most deaths occurred on Mondays and Tuesdays, with 809 deaths each, consistent with the higher volume of admissions on weekdays.

The mortality rate for weekday admissions was 10.47%, while weekend admissions showed a significantly higher mortality rate of 15.58% (Table 3). This resulted in a risk difference of 5.12%, a relative risk of 1.49, and an unadjusted odds ratio (OR) of 1.58 (95% CI: 1.48–1.68, *p* < 0.001), indicating that weekend admissions were associated with substantially increased odds of in-hospital death. A chi-square test (Table 3) confirmed that the difference in mortality rates between weekday and weekend admissions was statistically significant (χ^2^ = 148.55, *p* < 0.0001). The analysis of the confidence intervals also showed that the CIs did not overlap, which emphasises the reliability of these results. However, as these unadjusted analyses did not take into account potential confounding factors such as age, gender, and re-hospitalisation status, we subsequently performed adjusted analyses (Section 3.5).

The highest single-day mortality rate was observed on Sundays (15.74%), followed closely by Saturdays (15.42%) and Fridays (12.45%). These findings provide strong evidence of the weekend effect, where patients admitted on weekends face a heightened risk of in-hospital mortality (Figure 2).

### 3.3. Mortality by Year

Our analysis of annual mortality trends revealed variations in the magnitude of the weekend effect during the study period. The most considerable discrepancy was observed in 2021, when the mortality rate for weekend admissions was 8.0% higher than for weekday admissions (25.6% vs. 17.6%; *p* < 0.0001). This is the most pronounced weekend effect in the dataset.

Throughout the study period, the weekend effect worsened over time. Between 2010 and 2015, the difference in mortality between weekend and weekday admissions remained relatively stable, fluctuating between 3.8% and 5.4%. From 2016, however, the weekend effect increased, with the differences exceeding 5% annually. The weekend effect was particularly pronounced in 2017 (5.1%) and 2018 (4.3%).

A strong increase in the weekend effect was observed during the COVID-19 pan-demic (2020–2021) (Figure 3). In 2020, the difference in mortality rose to 6.7% and peaked at 8.0% in 2021. By 2022, the weekend effect decreased slightly to 5.6%, indicating a partial recovery of hospital operations after the pandemic, even though mortality differentials remained elevated compared to pre-pandemic levels (Table 4).

### 3.4. Unadjusted Analysis

To assess the raw association between weekend admissions and in-hospital mortality, we first calculated the unadjusted odds ratio (OR). The mortality rate for weekend admissions was 15.58%, compared with 10.47% for weekday admissions, an absolute increase of 48.81%. A chi-square test confirmed that this difference was statistically significant (χ^2^ = 148.55, *p* < 0.0001). The unadjusted odds ratio (OR) for mortality for weekend admissions compared to weekday admissions was OR = 1.58 (95% CI: 1.48–1.68, *p* < 0.001). This means that the mortality rate was 58% higher for patients admitted on weekends compared to those admitted on weekdays.

### 3.5. Adjusted Analysis

To evaluate the independent effects of the different predictors on in-hospital mortality, we conducted a multivariable logistic regression model adjusting for age, gender, re-hospitalisation status, and length of stay (LOS). The adjusted odds ratio (AOR) for mortality for weekend admissions compared to weekday admissions was AOR = 1.31 (95% CI: 1.22–1.41, *p* < 0.001). This means that even after controlling confounding factors, the mortality rate was 31% higher for patients admitted on weekends than for patients admitted on weekdays.

Age ≥ 65 years was the strongest predictor of mortality, with patients in this group having an almost threefold higher risk of death compared to younger patients (AOR = 2.88, 95% CI: 2.73–3.05, *p* < 0.001). In addition, male gender was significantly associated with increased mortality (AOR = 1.37, 95% CI: 1.28–1.47, *p* < 0.001). In contrast, re-hospitalisation status (AOR = 1.00, 95% CI: 0.96–1.05, *p* = 0.87) and length of stay (AOR = 1.05, 95% CI: 0.98–1.12, *p* = 0.18) were not statistically significant predictors. In other words, after adjusting for the various factors, there was no significant difference in the probability of death in the hospital between patients who were rehospitalised and those who were not. Although each additional day in hospital was associated with a 5% increase in the probability of in-hospital mortality, this association was not statistically significant.

To verify the robustness of our results, we applied a generalised linear model (GLM) with a complementary log–log (cloglog) link function. The results were consistent with those of the logistic regression analysis and confirmed that weekend admission remained a significant predictor of in-hospital mortality (adjusted odds ratio [AOR] = 1.30, 95% CI: 1.21–1.40, *p* < 0.001). This consistency between modelling approaches underlines the reliability of our results. Table 5 summarises the AORs and corresponding 95% confidence intervals for all major predictors included in the final model.

The statistical significance of our results and the non-overlapping confidence intervals emphasise the robust association between weekend admissions and increased mortality in hospital.

### 3.6. Interaction Analysis

To determine whether the weekend effect on mortality differed by age and gender, we included two interaction terms—weekend × age and weekend × gender—in the multivariable logistic regression model (Table 6). Neither interaction reached statistical significance (*p* = 0.312 for weekend × age and *p* = 0.215 for weekend × gender), indicating that the effect of weekend admission on in-hospital mortality was consistent across these demographic subgroups, suggesting that patient age and gender do not modify the association between weekend admission and mortality.

Although the interaction between weekend admission and age was not statistically significant, the stratified analysis revealed a clinically meaningful trend: the weekend effect was more pronounced among older patients (AOR = 1.60) compared to younger ones (AOR = 1.34). These patterns may guide future research and planning for age-specific weekend care strategies.

As a result, possible interventions or policy changes to mitigate this “weekend effect” should target hospital-wide processes rather than focusing on specific demographic groups. Future studies may explore other potential effect modifiers, such as comorbidity profiles or socioeconomic status, to further elucidate the nuances of the weekend effect.

### 3.7. Cause-Specific Mortality and Weekend Admission Patterns

Cardiovascular diseases were the main cause of hospital deaths, accounting for 70.4% of all deaths (3664/5205), followed by respiratory diseases (13.5%) and sepsis (5.1%) (Table 7). Patients who died from cardiovascular and respiratory diseases had an average age of about 77 years and a median length of stay (LOS) of 6.5–7 days. In contrast, patients with digestive system diseases and malignancies had a longer length of stay in hospital—over 8 days on average—before dying, while patients with cardiogenic shock had the shortest length of stay (3.7 days).

A comparison between weekday and weekend admissions showed that weekend admissions were disproportionately associated with deaths due to cardiovascular disease (*n* = 853), respiratory disease (*n* = 158), and sepsis (*n* = 59). Diseases of the digestive system (3.1% of all deaths) occurred at a younger mean age (64 years) but had one of the longest LOS (8.0 ± 7.5 days). Malignant diseases also had an above-average mean LOS (8.7 days). Less important cause of death categories included endocrine disorders (1.3%) and urogenital causes (0.9%), both reflecting an older age profile (mean ages of 75.7 and 78.5 years, respectively) and a moderate LOS (approximately 4–5 days). A small proportion of deaths fell into the “other” category (0.7%), which included conditions not represented in the main diagnostic groups.

We performed chi-square tests to determine whether each diagnosis-specific cause of death was disproportionately represented among weekend admissions. Although we observed more cardiovascular (*n* = 853), respiratory (*n* = 158), and septic (*n* = 59) deaths on weekends, these increases were not statistically significant compared to weekday deaths (all *p* > 0.05). Thus, no category of cause of death was found to be significantly associated with weekend admissions compared to weekdays.

## 4. Discussion

### 4.1. Overview of Findings

Weekend versus weekday admission is receiving increasing attention in healthcare research, with multiple studies reporting higher in-hospital mortality rates for weekend admissions—a phenomenon often termed the “weekend effect”. Weekend versus weekday admission is receiving increasing attention in healthcare research, with multiple studies reporting higher in-hospital mortality rates for weekend admissions—a phenomenon often termed the “weekend effect”. Motivated by these international findings and the potential implications for patient care, we sought to investigate this issue in our internal medicine ward.

Over the 13 years analysed, the overall in-hospital mortality rate for patients admitted on weekends was 15.58%, whereas weekday admissions had a lower mortality rate of 10.47%. Our study found an unadjusted OR of 1.58 (95% CI: 1.48–1.68, *p* < 0.001), meaning that weekend admissions had 58% higher mortality odds than weekday admissions. This is substantially greater than the pooled OR of 1.16 (95% CI: 1.10–1.23) reported in the most actual meta-analysis [5]. This suggests that the weekend effect is notably stronger in Slovakia than in many other healthcare systems. Even after adjustment for confounders, the weekend effect remained significant (AOR = 1.31, 95% CI: 1.22–1.41, *p* < 0.001), suggesting systemic factors likely contribute to the increased weekend mortality risk.

Our findings are consistent with the wider international evidence on the weekend effect [2,6,8,9]. Studies from Australia, the USA, and numerous European countries have shown that the results of weekend admissions are worse, especially in studies focusing on a single diagnosis, usually cardiovascular diseases such as acute myocardial infarction and aortic dissection [2,7,8,9,10]. While some studies suggest that controlling for disease severity and comorbidities may mitigate or explain some of the weekend effects, other studies report significant weekend-related differences [7,8,26]. In addition, large-scale analyses suggest that hospitals where residents work are particularly prone to increased weekend mortality [1]. As a large academic medical centre, our hospital frequently deals with complex patient cases. It relies on clinicians in training, which can exacerbate the differences between weekend and weekday treatment processes and outcomes.

### 4.2. Interaction Analysis Findings

Our analysis of interaction terms (weekend × age and weekend × sex) indicated that neither patient age nor sex significantly influenced the weekend effect (*p* = 0.312 and *p* = 0.215, respectively). This consistency across subgroups suggests that the increased mortality risk associated with weekend admissions is not driven by demographic differences. Rather, it points to a system-level issue—likely related to factors such as staffing levels, resource availability, and compliance with care protocols—affecting all patients equally regardless of age or sex.

### 4.3. Interpreation of Cause-Specific Mortality Patterns

Our cause-specific mortality analysis shows which diseases are most affected by weekend-related supply shortages. Cardiovascular diseases were the main cause of hospital deaths (70.4%), followed by respiratory diseases and sepsis.

In patients with cardiogenic shock, the length of stay in hospital before death was particularly short (3.7 days on average), which emphasises the rapid progression and life-threatening nature of this condition. In such cases, even short delays in treatment can be fatal, and these delays may occur more frequently at weekends due to reduced staffing levels or limited availability of specialist physicians.

In contrast, conditions such as digestive disorders and malignancies were associated with longer hospital stays, which may reflect the need for longer diagnostic and therapeutic approaches. Overall, the weekend effect was most pronounced for cardiovascular, respiratory, and sepsis-related deaths. This suggests that time-critical illnesses are particularly vulnerable to interruptions in healthcare that may occur at weekends [33].

### 4.4. Temporal Trends and Association with the COVID-19 Pandemic

Our analysis revealed that the magnitude of the weekend effect fluctuated over time, with a notable worsening trend after 2016. The most pronounced disparities were observed during the COVID-19 pandemic in 2020 and 2021, where the weekend effect peaked at 8.0% in 2021. These findings indicate that the strain on healthcare resources during the pandemic may have exacerbated pre-existing inefficiencies, contributing to higher weekend mortality rates [38].

Moreover, our internal department was designated as a “red hospital” during the COVID-19 pandemic, signifying its role as a primary centre for treating COVID-19 patients. This designation likely intensified challenges during weekend admissions as the influx of COVID-19 patients further strained hospital resources. Even in the post-pandemic period (2022), the weekend effect remained elevated, suggesting that disruptions to hospital workflows and persistent staffing shortages had lasting consequences.

### 4.5. Potential Mechanisms Behind the Weekend Effect

Despite numerous investigations, there is no universal consensus on the magnitude of the weekend effect or the precise mechanisms driving it. Several hypotheses have been proposed to explain this phenomenon. One possible factor is reduced hospital staffing during weekends, which may lead to delays in diagnostics, interventions, and overall patient care [9]. Studies suggest that lower physician-to-patient ratios and limited availability of specialised consultants on weekends contribute to worse patient outcomes [9,10]. Another hypothesis is that physicians working on weekends may have less experience, potentially impacting clinical decision-making. However, research adjusting for physician experience levels has shown that mortality differences persist even after accounting for this factor [12].

Additionally, retrospective analyses indicate that patients admitted on weekends often have higher risk profiles and greater severity of illness compared to weekday admissions, potentially contributing to increased mortality [5]. Furthermore, a 2016 study suggested that mortality disparities may not be exclusively limited to weekends but could also be influenced by overall variations in healthcare quality and resource allocation at different times of the day and week [25]. These challenges may also be relevant within the Slovak healthcare system. According to the INESS (Institute of Economic and Social Studies) report [34], the sector is characterised by certain rigidities, which could contribute to variations in care delivery and the weekend effect.

### 4.6. Strengths and Limitations

This study benefits from a robust dataset spanning 13 years and more than 45,000 admissions, which provides considerable statistical power and allows for in-depth analysis of trends over time. The use of multivariable logistic regression modelling enabled the adjustment of key confounding factors, thereby strengthening the reliability of the observed associations. In addition, the cause-specific mortality analysis identified the clinical conditions most affected by weekend admissions, providing insight into potential targets for intervention. Importantly, by delivering the first long-term Slovak evidence on the weekend effect, the study fills a critical regional evidence gap that has hindered meaningful cross-country comparisons within Central and Eastern Europe.

Despite these strengths, the study also has its limitations. First, the data are from a single large academic referral centre, which may limit the generalisability of the results to smaller community hospitals or other healthcare facilities. Diagnosis-specific analyses relied on ICD codes, which may be incomplete or inaccurate if comorbidities or secondary diagnoses are not fully documented—a common limitation of administrative data. In addition, the study did not include detailed clinical process data such as time to intervention or availability of specialists (e.g., cardiology services), which limits our understanding of the mechanisms behind the increased cardiovascular mortality observed at weekends. Future research incorporating time-stamped intervention data and service availability metrics could help to clarify these structural factors.

We did not include national public holidays as a separate category of “out-of-hours,” although they may reflect the challenges associated with weekends in healthcare. Also, mortality data were limited to in-hospital events; deaths that occurred after discharge were not included. This could underestimate the true risk of death—particularly for patients discharged early—and lead to a bias in estimating the impact of admission timing. Although we included length of stay in our modelling, future studies that take into account post-discharge outcomes would provide a more comprehensive picture.

Furthermore, our dataset lacked a distinction between elective and emergency admissions, so analyses to assess whether the weekend effect was actually due to temporal differences or simply a different case mix were not possible. Finally, there is the possibility that unmeasured variables such as socioeconomic status or detailed comorbidity indices may confound the results, emphasising the need for more detailed and comprehensive data for future studies.

## 5. Conclusions

Based on a comprehensive literature search, our study provides the first comprehensive analysis of the effect of the weekend in Slovakia and confirms a higher hospital mortality rate for weekend admissions. Even after adjusting for key confounders, weekend admission remained an independent predictor of mortality, with cardiovascular, respiratory, and sepsis-related deaths being the most affected. The observed weekend effect in Slovakia (AOR = 1.31, 95% CI: 1.22–1.41) is stronger than the pooled estimate from the BMJ Open 2019 meta-analysis (OR = 1.16, 95% CI: 1.10–1.23 across all admission types). The observed trends, particularly during the COVID-19 pandemic, highlight the importance of understanding the factors contributing to weekend-related variations in mortality. As the weekend effect is likely multifactorial, future research should identify specific factors at the hospital and system level that contribute to increased weekend mortality. One possible strategy to mitigate the weekend effect could be to improve access to high-level decision-makers and expand the availability of key diagnostic and therapeutic services on weekends. Such structural adjustments have been associated with better outcomes in other healthcare systems and could be adapted to the Slovakian context. A deeper understanding of these factors is necessary to develop strategies that ensure equitable and high-quality care regardless of the day of admission.

## Figures and Tables

**Figure 1 healthcare-13-01412-f001:**
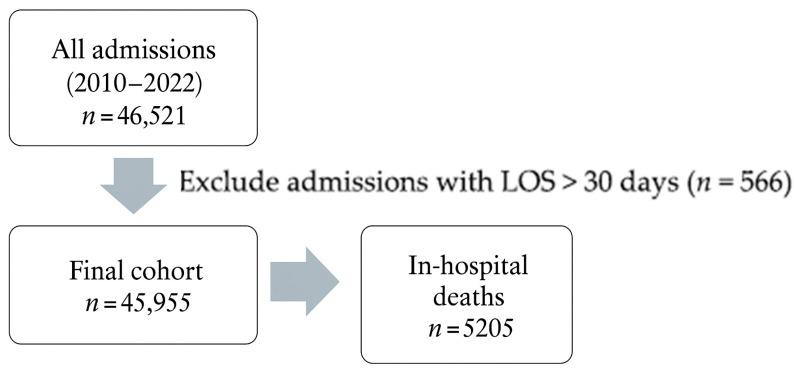
Flowchart of exclusion criteria and final cohort.

**Figure 2 healthcare-13-01412-f002:**
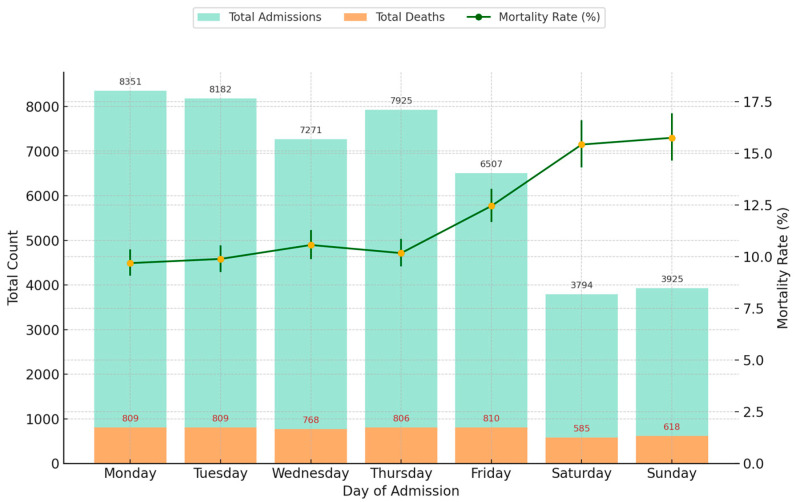
Total hospital admissions, in-hospital deaths, and mortality rates by day of admission (2010–2022) with 95% CI.

**Figure 3 healthcare-13-01412-f003:**
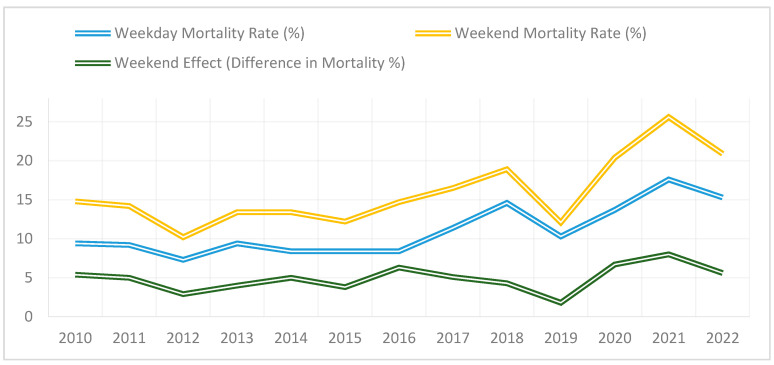
Trends in weekday vs. weekend mortality rates 2010–2022.

**Table 1 healthcare-13-01412-t001:** Comparison of baseline characteristics between weekday and weekend admissions.

Variable	Weekday Admissions	Weekend Admissions	*p*-Value
Age (mean)	68.77	69.96	<0.0001
Sex (female)	19,632	4035	—
Sex (male)	18,604	3684	0.139

**Table 2 healthcare-13-01412-t002:** Descriptive characteristics of the final cohort of hospital admissions and in-hospital mortality by day of admission (2010–2022).

	**All Hospitalisations** **(Within 30 Days of Admission)**	**Males, *n* (%)**	**Females, *n* (%)**
Total (*n*, %)	45,955	22,288 (48.5)	23,667 (51.5)
Age, mean ± SD, (years) Median [IQR]	69.0 ± 16.2 72.1 [60.2–81.2]	66.1 ± 15.9 66.1 [56.8–78.4]	71.6 ± 16.0 70.4 [63.4–81.9]
LOS, mean ± SD, (days) Median [IQR]	6.1 ± 5.2 5 [2–8]	6.1 ± 5.3 5 [2–8]	6.1 ± 5.1 5 [2–8]
Mon	8351 (18.2)	4039 (18.1)	4312 (18.2)
Tue	8182 (17.8)	4049 (18.2)	4133 (17.5)
Wed	7271 (15.8)	3536 (15.9)	3735 (15.8)
Thu	7925 (17.2)	3894 (17.5)	4031 (17.0)
Fri	6507 (14.2)	3086 (13.8)	3421 (14.5)
Subtotal for Weekdays	38,236	18,604	19,632
Sat	3794 (8.3)	1772 (8.0)	2022 (8.5)
Sun	3925 (8.5)	1912 (8.6)	2013 (8.5)
Subtotal for Weekend	7719	3684	4035
	**In-Hospital Deaths** **(Within 30 Days of Admission)**	**Males, *n* (%)**	**Females, *n* (%)**
Total (*n*, %)	5205	2491 (47.9)	2714 (52.1)
Age, mean ± SD, (years) Median [IQR]	77.0 ± 11.8 79.2 [70.1–85.8]	74.6 ± 12.0 76.0 [67.1–83.5]	79.2 ± 11.2 81.6 [73.2–87.1]
LOS, mean ± SD, (days) Median [IQR]	6.5 ± 6.6 4 [1–10]	6.7 ± 6.7 4 [1–10]	6.4 ± 6.6 4 [1–10]
Mon	809 (15.5)	378 (15.2)	431 (15.9)
Tue	809 (15.5)	383 (15.4)	426 (15.7)
Wed	768 (14.8)	390 (15.7)	378 (13.9)
Thu	806 (15.5)	385 (15.5)	421 (15.5)
Fri	810 (15.6)	373 (15.0)	437 (16.1)
Subtotal for Weekdays	4002	1909	2093
Sat	585 (11.2)	298 (12.0)	287 (10.6)
Sun	618 (11.9)	284 (11.4)	334 (12.3)
Subtotal for Weekend	1203	582	621

**Table 3 healthcare-13-01412-t003:** Comparison of in-hospital mortality between weekday and weekend admissions, including risk difference, relative risk, and odds ratio.

Admission Type	Weekday Admissions	Weekend Admissions	Total
Total admissions (*n*)	38,236	7719	45,955
In-hospital deaths (*n*)	4002	1203	5205
Mortality rate (%)	10.47%	15.58%	11.32%
Chi-square (χ^2^)	148.55		
*p*-value	<0.0001		
Risk difference	+5.12		
Relative risk	1.49		
Odds ratio (unadjusted)	1.58		

**Table 4 healthcare-13-01412-t004:** Annual mortality trends.

Year	Weekday Admissions (*n*)	Weekday Deaths (*n*)	Weekday Mortality Rate (%)	95% CI	Weekend Admissions (*n*)	Weekend Deaths (*n*)	Weekend Mortality Rate (%)	95% CI
2010	3077	289	9.4%	8.41–10.47%	596	88	14.8%	12.14–17.84%
2011	3228	298	9.2%	8.28–10.28%	636	90	14.2%	11.66–17.08%
2012	3377	248	7.3%	6.51–8.27%	618	63	10.2%	8.05–12.83%
2013	3508	330	9.4%	8.48–10.42%	658	88	13.4%	10.98–16.19%
2014	3529	296	8.4%	7.52–9.35%	689	92	13.4%	11.01–16.10%
2015	3517	294	8.4%	7.50–9.31%	698	85	12.2%	9.96–14.83%
2016	3276	275	8.4%	7.45–9.44%	604	89	14.7%	12.24–17.66%
2017	3251	370	11.4%	10.30–12.55%	618	102	16.5%	13.80–19.60%
2018	2336	340	14.6%	13.08–16.16%	482	91	18.9%	15.47–22.83%
2019	2891	299	10.3%	9.22–11.60%	569	69	12.1%	9.57–15.22%
2020	2263	310	13.7%	12.28–15.25%	520	106	20.4%	17.08–24.10%
2021	1934	340	17.6%	15.93–19.38%	519	133	25.6%	22.01–29.60%
2022	2041	313	15.3%	13.77–17.03%	511	107	20.9%	17.66–24.60%

**Table 5 healthcare-13-01412-t005:** Adjusted odds ratios (AORs) for mortality predictors and statistical significance.

Factor	AOR (Adjusted Odds Ratio)	Lower CI	Upper CI	Statistical Significance (*p*-Value)
Weekend admission	1.31	1.22	1.41	<0.001
Age ≥ 65	2.88	2.73	3.05	<0.001
Male gender	1.37	1.28	1.47	<0.001
Re-hospitalisation	1	0.96	1.05	0.87
Length of stay	1.05	0.98	1.12	0.18

**Table 6 healthcare-13-01412-t006:** Interaction analysis of weekend admission with age and gender for in-hospital mortality.

Interaction Term	AOR	Lower CI	Upper CI	*p*-Value
Weekend × Age	1.02	0.97	1.07	0.312
Age < 65 (stratified)	1.34	-	-	-
Age ≥ 65 (stratified)	1.60	-	-	-
Weekend × Gender	1.05	0.99	1.11	0.215

**Table 7 healthcare-13-01412-t007:** Cause-specific mortality and weekend admission patterns.

Cause of Death	*n* (%)	Male, *n* (%)	Age, Mean ± SD (Years)	LOS ± SD (Days)	Weekday Deaths (*n*)	Weekend Deaths (*n*)	*p*-Value
Total deaths	5205	2491	77.0 ± 11.8	6.5 ± 6.6	4002	1203	-
Cardiovascular	3664 (70.4)	1734 (69.6)	77.0 ± 11.3	6.5 ± 6.6	2811 (70.2)	853 (70.9)	0.683
Respiratory	701 (13.5)	332 (13.3)	77.9 ± 11.6	7.0 ± 6.8	543 (13.6)	158 (13.1)	0.735
Sepsis and other infectious	265 (5.1)	137 (5.5)	77.4 ± 11.0	6.7 ± 6.8	205 (5.1)	59 (4.9)	0.820
Digestive	162 (3.1)	104 (4.2)	64.0 ± 13.1	8.0 ± 7.5	129 (3.2)	33 (2.7)	0.455
Cardiogenic shock	138 (2.7)	74 (3.0)	73.9 ± 14.3	3.7 ± 5.3	105 (2.6)	33 (2.7)	0.901
Malignancy	121 (2.3)	47 (1.9)	71.7 ± 11.6	8.7 ± 7.8	94 (2.3)	28 (2.3)	0.968
Endocrinous	69 (1.3)	27 (1.1)	75.7 ± 11.4	4.6 ± 5.2	47 (1.2)	22 (1.8)	0.301
Urogenital	45 (0.9)	21 (0.8)	78.5 ± 9.5	5.2 ± 5.2	37 (0.9)	8 (0.7)	0.106
Other	40 (0.7)	15 (0.6)	73.0 ± 13.7	6.1 ± 5.2	30 (0.7)	0 (0)	0.011

## Data Availability

The raw data supporting the conclusions of this article will be made available by the authors on request.

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
