# Peer review of "The Weekend Effect on In-Hospital Mortality—First 13-Year Retrospective Observational Study in Slovakia"

_healthcare, 2025, doi:10.3390/healthcare13121412_

Round 1

Reviewer 1 Report

Comments and Suggestions for Authors

Dear authors,

This is very informative research. I would like to suggest minor revison. Would you like to consider these suggestion wich might increase a quality of your paper.

Title

  • I would suggest shortening the title and removing the phrase “Central Europe”, as the study is based on data from a single hospital in one country.

Introduction

  • I wouldn’t begin with numbers right away. I suggest slightly expanding the introduction to capture the reader’s interest first.
  • The details about the hospital should be moved to the Methods section, preferably into a dedicated subsection.

Methods

  • I would always include a clear definition of “hospitalisation.” For example, does it include day cases, observation-only stays, etc.?
  • The sentence “For this study, we focused on patients who died or were discharged within 30 days of admission” should clearly state that it refers to patients who died during their hospital stay.
  • Which edition of the ICD-10 was used? Please provide a citation or reference to the exact version.

Results

  • The distribution of length of stay (LOS) is not normal. I recommend reporting the median and interquartile range (IQR) alongside the mean and standard deviation.
  • Table 1 is missing percentages for days of the week.
  • Please rephrase: “ended in 30 days” → better as “within 30 days of admission.”
  • In Figure 1, a value is missing after the equals sign (“=”).
  • You state that the highest number of deaths occurred on Tuesdays and Mondays, but the actual highest number appears to be on Fridays. This paragraph should be placed above the table, with a clear reference to the table.
  • Since you already used R, consider producing the plots in R as well. The current plots appear to be generated in Excel and are not publication quality. Please clean up Figure 2. I suggest adding values on top of bars, and using points to clarify what the “rate” refers to.
  • You have two figures labeled as Figure 2. Do the two lines represent confidence intervals or something else? Please clarify. All authors should read the full manuscript carefully — such mistakes would likely have been caught if the entire team reviewed the documen
  • If I understood correctly, you performed analyses based on the cause of death. Why didn’t you also conduct analyses based on the primary cause of hospitalisation? That might help identify specific reasons for the increased weekend mortality.

Discussion

  • Could the observed effect be explained by the fact that weekday admissions (Monday–Friday) are more often planned, while weekend (Saturday–Sunday) admissions are mostly unplanned or emergencies?
  • Are there similar studies from comparable countries (e.g., Poland, Czech Republic, Hungary, Romania, Austria)?
  • The authors have been commendably transparent in presenting the limitations of the study.
  • In future work, consider including national holidays as part of the “weekend” category.
  • Please be cautious: observational studies do not provide causal inference — they show associations, not causes.
  • If you discuss organisational characteristics as possible contributors to the weekend effect, I suggest including at least one concrete recommendation for improving hospital organisation or weekend staffing.

Author Response

Response to Reviewer 1 Comments

1. Summary

Thank you for taking the time to review our manuscript. Detailed responses are provided below, and the corresponding revisions and corrections are highlighted in red in the resubmitted file (article).

2. Questions for General Evaluation

Reviewer’s Evaluation

Response and Revisions

Does the introduction provide sufficient background and include all relevant references?

Yes/Can be improved/Must be improved/Not applicable

[Please give your response if necessary. Or you can also give your corresponding response in the point-by-point response letter. The same as below]

Are all the cited references relevant to the research?

Yes/Can be improved/Must be improved/Not applicable

Is the research design appropriate?

Yes/Can be improved/Must be improved/Not applicable

Are the methods adequately described?

Yes/Can be improved/Must be improved/Not applicable

Are the results clearly presented?

Yes/Can be improved/Must be improved/Not applicable

Are the conclusions supported by the results?

Yes/Can be improved/Must be improved/Not applicable

3. Point-by-point response to Comments and Suggestions for Authors

Comment 1: I would suggest shortening the title and removing the phrase “Central Europe”, as the study is based on data from a single hospital in one country.

Response 1: We appreciate this helpful suggestion. We have revised the title and removed the phrase “Central Europe” to better reflect the scope of the study. The updated title is:

“The Weekend Effect on In-Hospital Mortality: A 13-Year Retrospective Observational Study from Slovakia.”

Comment 2: I wouldn’t begin with numbers right away. I suggest slightly expanding the introduction to capture the reader’s interest first.

Response 2: Many thanks for this helpful suggestion. We have revised the introduction to provide a broader context before presenting numerical data. The updated version introduces the weekend effect as a global phenomenon and emphasises the existing evidence gap in Central and Eastern Europe, particularly Slovakia. These changes aim to better engage the reader and emphasise the relevance of our study from the outset. The revised introduction can be found on page 2, lines 38-77 of the revised manuscript.

Comments 3-6:

·         The details about the hospital should be moved to the Methods section, preferably into a dedicated subsection.

·         I would always include a clear definition of “hospitalisation.” For example, does it include day cases, observation-only stays, etc.?

·         The sentence “For this study, we focused on patients who died or were discharged within 30 days of admission” should clearly state that it refers to patients who died during their hospital stay.

·         Which edition of the ICD-10 was used? Please provide a citation or reference to the exact version.

Response 3-6: Thank you for these suggestions. We have made the following revisions to the Methods section to enhance clarity and methodological transparency:

·         Details describing the hospital and its setting have been moved to subsection titled “Study Setting and Data Sources.”

·         We clarified the definition of hospitalisation by stating that the analysis included only inpatient admissions with a length of stay up to 30 days, and that day cases and observation-only stays were excluded.

·         The description of the outcome variable was revised to explicitly state that we measured in-hospital mortality within 30 days, i.e., deaths occurring during the hospital stay.

·         We also specified that diagnostic coding was based on the 10th Revision of the International Classification of Diseases (ICD-10), referencing the WHO 2016 version.

These changes can be found on page 2-3, lines 78-115 of the revised manuscript.

Comment 7: The distribution of length of stay (LOS) is not normal. I recommend reporting the median and interquartile range (IQR) alongside the mean and standard deviation.
Response 7: Thank you for this valuable suggestion. We agree that the distribution of length of stay (LOS) does not follow a normal distribution. Accordingly, we have revised the manuscript to include the median and interquartile range (IQR) alongside the mean and standard deviation for LOS. This provides a more comprehensive summary of the data distribution. The updated descriptive statistics can be found in 3.1. Patient Characteristics and Admission Patterns part of th e revised manuscript.

Comment 8-9: Table 1 is missing percentages for days of the week. Please rephrase: “ended in 30 days” → better as “within 30 days of admission.”
Responses 8-9: Table 1 is now corrected on page 4. + also we rephrased the sentence

Comment 10: In Figure 1, a value is missing after the equals sign (“=”).
Response 10: Figure 1 is now corrected on page 5 (n=566)

Comment 11: You state that the highest number of deaths occurred on Tuesdays and Mondays, but the actual highest number appears to be on Fridays. This paragraph should be placed above the table, with a clear reference to the table.

Response 11: Thank you for pointing this out. We have corrected the statement to accurately reflect that the highest number of deaths occurred on Fridays. Additionally, we have moved the paragraph as suggested so that it now appears above the table, with a clear reference directing readers to the table for supporting data (Table 1.)

Comments 12-13: Since you already used R, consider producing the plots in R as well. The current plots appear to be generated in Excel and are not publication quality. Please clean up Figure 2. I suggest adding values on top of bars, and using points to clarify what the “rate” refers to.

You have two figures labeled as Figure 2. Do the two lines represent confidence intervals or something else? Please clarify. All authors should read the full manuscript carefully — such mistakes would likely have been caught if the entire team reviewed the document

Responses 12-13: Thank you for this valuable feedback. In response, we have recreated the figure in R to ensure higher visual and publication quality. Figure 2 has been revised accordingly: we added values on top of the bars and included points to better clarify the “rate” being presented. We also corrected the duplicate figure numbering—there is now a single Figure 2 with consistent labeling throughout the manuscript. Additionally, the lines in the graph have been clarified in the caption to indicate that they represent 95% confidence intervals (according to another reviewer).

Comment 14: If I understood correctly, you performed analyses based on the cause of death. Why didn’t you also conduct analyses based on the primary cause of hospitalisation? That might help identify specific reasons for the increased weekend mortality.

Responses 14: Thank you for this valuable observation. Our analysis focused on the cause of death rather than the primary reason for hospitalisation because our primary objective was to identify the clinical conditions directly associated with in-hospital mortality. This focus aligns with previous research (e.g., DOI: 10.1111/imj.15723).

In addition, in our previous workflow we reviewed and compared all four available diagnosis fields in our dataset—admission diagnosis, primary diagnosis, main diagnosis, and cause of death—and decided that we will further look at the cause of death, which offeres the most direct and clinically relevant insight into the reasons of in-hospital deaths. This decision was made to enhance the interpretability of mortality patterns and focus on outcomes most relevant to patient safety and quality of care.

Comment 15: Could the observed effect be explained by the fact that weekday admissions (Monday–Friday) are more often planned, while weekend (Saturday–Sunday) admissions are mostly unplanned or emergencies?
Response 15: The study was conducted at the 5th Department of Internal Medicine in Bratislava, which primarily admits acute patients from the local region, but also offers elective hospitalisations were included. So, the dataset includes a mix of both types of admissions.

Comment 16:  Are there similar studies from comparable countries (e.g., Poland, Czech Republic, Hungary, Romania, Austria)?
Response 16: Yes, thank you for highlighting these countries. All further studies are cited in the manuscript now (citations 23-25)
https://pubmed.ncbi.nlm.nih.gov/33784248/ Hungary
https://pubmed.ncbi.nlm.nih.gov/33005270/ Poland
https://pmc.ncbi.nlm.nih.gov/articles/PMC11193957/ Romania

Comment 17: The authors have been commendably transparent in presenting the limitations of the study.

Response 17: We sincerely thank you for acknowledging our effort to provide a clear and transparent discussion of the study’s limitations. We believe that openly addressing methodological and contextual constraints strengthens the validity and reliability of our findings.

Comment 18:  In future work, consider including national holidays as part of the “weekend” category.

Response 18: Thank you for this thoughtful suggestion. We agree that national holidays may share similar operational constraints with weekends, such as reduced staffing and limited services. While our current dataset did not specifically flag holidays, we acknowledge this as an important consideration for future studies and will aim to incorporate this variable in follow-up analyses. (correction line 466-472)

Comment 19: Please be cautious: observational studies do not provide causal inference — they show associations, not causes.

Response 19: We fully agree and appreciate the reviewer’s important reminder. We have reviewed the manuscript once again to ensure that our language reflects the observational nature of the study. Where applicable, we have revised phrasing to clarify that our findings indicate associations rather than causal relationships.

Comment 20: If you discuss organisational characteristics as possible contributors to the weekend effect, I suggest including at least one concrete recommendation for improving hospital organisation or weekend staffing.

Response 20: Thank you for encouraging us to make our article more actionable. In response, we have added a concrete recommendation. We agree that organizational factors likely play a role in the weekend effect. However, given the observational nature of our study and the absence of direct data on staffing levels, resource availability, or clinical workflows, we prefer not to make prescriptive recommendations. Instead, we highlight the need for future research that directly investigates these operational aspects to support evidence-based improvements in hospital organization and weekend care delivery.

Reviewer 2 Report

Comments and Suggestions for Authors

This manuscript addresses an important and underexplored area in hospital-based epidemiology in Central Europe, specifically assessing the “weekend effect” in Slovakia using a large retrospective dataset. The study is well-structured, and the findings are both statistically and clinically significant. However, several methodological clarifications and improvements are required to enhance the manuscript’s rigor, transparency, and potential impact.
- The study fills a crucial gap in the literature by providing evidence on the weekend effect from a Central European context, which has been underrepresented in prior global studies. The manuscript would benefit from a slightly expanded discussion section comparing results with similar studies in other European countries to contextualize the observed 31% increased mortality.
- Please clarify the inclusion and exclusion criteria for the analyzed hospitalizations. Were any specific patient populations (e.g., palliative care, trauma cases) excluded from the analysis?
- The rationale for choosing logistic regression over other survival models (e.g., Cox proportional hazards) should be justified, particularly when accounting for variable lengths of stay.
- It would be helpful to report whether weekend admissions differed significantly from weekday admissions in terms of baseline characteristics (e.g., age, diagnosis distribution), as these could be confounding variables.
- The definition of the weekend as Saturday–Sunday is conventional, but was there consideration of Friday evening or holiday periods, which in some literature are also associated with increased risk? A sensitivity analysis including these timeframes could strengthen the conclusions.
- The manuscript notes a peak during the COVID-19 pandemic but does not explore this in depth. Please consider stratifying the analysis pre- and post-pandemic to better understand temporal shifts and potential healthcare system strain effects.
- The identification of cardiovascular diseases as the leading cause of weekend deaths is an important observation. Additional details on these cases (e.g., time to intervention, availability of cardiology services on weekends) would provide valuable insight into the mechanisms of the weekend effect.
- Please ensure consistency in terminology: use either “weekend effect” or “weekend admission mortality” throughout.
- Consider improving the abstract’s clarity by simplifying overly technical language for better accessibility to a broader readership.

Author Response

Response to Reviewer 2 Comments

1. Summary

Thank you for taking the time to review our manuscript. We have provided detailed responses below, and all corresponding revisions and corrections are highlighted in red in the resubmitted article.

2. Questions for General Evaluation

Reviewer’s Evaluation

Response and Revisions

Does the introduction provide sufficient background and include all relevant references?

Yes/Can be improved/Must be improved/Not applicable

[Please give your response if necessary. Or you can also give your corresponding response in the point-by-point response letter. The same as below]

Are all the cited references relevant to the research?

Yes/Can be improved/Must be improved/Not applicable

Is the research design appropriate?

Yes/Can be improved/Must be improved/Not applicable

Are the methods adequately described?

Yes/Can be improved/Must be improved/Not applicable

Are the results clearly presented?

Yes/Can be improved/Must be improved/Not applicable

Are the conclusions supported by the results?

Yes/Can be improved/Must be improved/Not applicable

3. Point-by-point response to Comments and Suggestions for Authors

Comment 1: This manuscript addresses an important and underexplored area in hospital-based epidemiology in Central Europe, specifically assessing the “weekend effect” in Slovakia using a large retrospective dataset. The study is well-structured, and the findings are both statistically and clinically significant. However, several methodological clarifications and improvements are required to enhance the manuscript’s rigor, transparency, and potential impact.

- The study fills a crucial gap in the literature by providing evidence on the weekend effect from a Central European context, which has been underrepresented in prior global studies. The manuscript would benefit from a slightly expanded discussion section comparing results with similar studies in other European countries to contextualize the observed 31% increased mortality.

- Please clarify the inclusion and exclusion criteria for the analyzed hospitalizations. Were any specific patient populations (e.g., palliative care, trauma cases) excluded from the analysis?

Response 1: Thank you for your thoughtful comment and for acknowledging the relevance of our study. We agree that contextualizing our findings within the broader European literature enhances the manuscript’s value. In response, we have expanded the Discussion section to include a comparison with studies from other European countries, highlighting both consistencies and regional differences in the magnitude of the weekend effect.

Regarding the methodology, we have now clarified the inclusion and exclusion criteria in the Methods section. Specifically, we included all hospitalisations with a length of stay of 30 days or less, irrespective of diagnosis, age, or admission type. Admissions exceeding 30 days were excluded to maintain consistency with our focus on acute care outcomes. No specific diagnostic groups (e.g., palliative care, trauma cases) were excluded, as our aim was to capture a comprehensive view of hospital mortality patterns. This clarification has been added to the revised manuscript (lines 93-99).

Comment 2: The rationale for choosing logistic regression over other survival models (e.g., Cox proportional hazards) should be justified, particularly when accounting for variable lengths of stay.

Response 2: Thank you for this important observation. Survival analysis is indeed a powerful tool for modeling time-to-event data, especially when examining the timing of events such as death and accounting for censoring (e.g., patients discharged alive). However, our study focused on a binary outcome — in-hospital mortality within a fixed 30-day observation window — and excluded stays longer than 30 days. Given this design, logistic regression was an appropriate and statistically valid choice to estimate the odds of death within this period.

Additionally, our dataset did not contain precise timestamps for discharge or death — only admission dates and outcomes. This limited our ability to model exact survival times, which are essential for techniques like the Cox proportional hazards model. Instead, we adjusted for length of stay (LOS) within the logistic regression framework, allowing us to account for variability in hospitalisation duration.

While survival analysis could provide further insight into the timing of death relative to admission, our approach prioritised estimating the overall risk of in-hospital mortality within a defined timeframe. In future research, with access to more granular time-to-event data, we would consider incorporating survival models as a complementary analytic method.

Comment 3: It would be helpful to report whether weekend admissions differed significantly from weekday admissions in terms of baseline characteristics (e.g., age, diagnosis distribution), as these could be confounding variables.
Response 3: Thank you for this valuable suggestion. We agree that differences in baseline characteristics could represent potential confounders. Accordingly, we have added a new subsection within the Results (Section 3.1) presenting a detailed comparison of age and gender between weekday and weekend admissions. This includes statistical testing showing that patients admitted on weekends were significantly older than those admitted on weekdays (p < 0.0001), while gender distribution was not significantly different (p = 0.139). Furthermore, we compared the distribution of diagnoses among deaths between the two groups and found no statistically significant difference (p = 0.615), suggesting comparable case-mix. These additions are intended to improve transparency and help contextualize the observed weekend effect. The new content can be found in Section 3.1 (lines 188-209), with the associated table labeled as Table 2.

Comment 4:  The definition of the weekend as Saturday–Sunday is conventional, but was there consideration of Friday evening or holiday periods, which in some literature are also associated with increased risk? A sensitivity analysis including these timeframes could strengthen the conclusions.

Response 4: Thank you for this thoughtful suggestion. We defined the weekend as Saturday and Sunday in line with the conventional classification used in much of the existing literature on the weekend effect. While we recognize that some studies have extended this definition to include Friday evenings or public holidays due to similar patterns in hospital staffing and service availability, our study was limited to the available date-based admission data, which did not include time-of-day or explicit holiday flags.

We agree that exploring these additional timeframes could provide further insight and represent a valuable direction for future research, particularly if more detailed temporal data becomes available.

Comment 5: The manuscript notes a peak during the COVID-19 pandemic but does not explore this in depth. Please consider stratifying the analysis pre- and post-pandemic to better understand temporal shifts and potential healthcare system strain effects.

Response 5: We agree that the COVID-19 pandemic had a profound impact on hospital operations and patient outcomes, and that a more detailed temporal analysis could provide additional insight into how healthcare system strain may have influenced the weekend effect.

While our manuscript highlights a visible peak in weekend mortality during the pandemic years (particularly 2020–2021), the scope of this study was not designed to explore pandemic-specific effects in depth. Our primary objective was to assess the overall presence and consistency of the weekend effect across a 13-year period.

We acknowledge that stratifying the analysis into pre- and post-pandemic periods would be a meaningful extension, particularly in identifying system-level disruptions. We consider this a valuable avenue for future investigation, especially with additional resources or expanded datasets.

Comment 6: The identification of cardiovascular diseases as the leading cause of weekend deaths is an important observation. Additional details on these cases (e.g., time to intervention, availability of cardiology services on weekends) would provide valuable insight into the mechanisms of the weekend effect.

Response 6: hank you for emphasising this important point. We agree that further details such as time to intervention or availability of cardiology services would provide valuable insights into the mechanisms behind the weekend effect. However, our current dataset did not include precise timing of clinical interventions or availability of services on different days. This represents an important limitation of the study, which we now address in the Discussion section. We also emphasise that this is a key area for future research to uncover structural or procedural factors for the increased cardiovascular mortality on weekends.

Comment 7:  Please ensure consistency in terminology: use either “weekend effect” or “weekend admission mortality” throughout.

Response 7: Thank you for this helpful suggestion. We have carefully reviewed the manuscript and ensured consistent terminology throughout.

Comment 8: Consider improving the abstract’s clarity by simplifying overly technical language for better accessibility to a broader readership.

Response 8: Thank you for the suggestion. We have revised the summary to simplify the language and improve clarity so that it is more accessible to a wider readership while maintaining the scientific integrity of the findings.

Reviewer 3 Report

Comments and Suggestions for Authors

Dear Authors, I read with interest your manuscript.

Here are my concerns:

1) - lines 54-60 are insufficient for providing a meaningful context. The geographic variability of the weekend phenomenon is not quantitavely reported, so that the reader only understands that it is widespread. I suggest the authors to provide some detailed figures. 

2) - lines 65-70 are describing the settings, I suggest to move them into the materials/methods section.

3) In the introduction is not decribed why the weekend phenomena is important in healthcare. Even though it may be obvious that mortality is not generally a good outcome, the patient perspective is very different from the facility one. This may be useful in highlighting the possible knowledge gap. If it is reputed a global phenomenon, the fact that in Slovakia it is not yet described appear as a poor rationale. If, as cited in line 60, the health care system plays a crucial role, some context should be provided. 

4) I suggest to add a methods section called "data extraction" after "study objectives" and report the relative information, which are not clear from lines 83 and foll.

5) line 143, describing an exclusion criterion, should be moved in matherials/method section.

6) lines 146-150 are not clear:  if an exact time is not available, how can be used date/time criteria (Mon 00:00 to Fri 23:59)? Please explain.

7) Please fix Fig.1 text (n=

8) line 166 - line 98 - the authors name the outcome "in hospital mortality". This deserves a methodological explaining: is the death event noticed only if the patient dies in the same hospital?  If so, the longer stays are at higher risk of death only because are not censored. Even if the logistic model controlled for LOS, I suggest to discuss the possible implication of not knowing if the death event happened when discharged. 

9) lines 229-231 are not clear. Please fix this.

10) maybe the ethical considerations deserve a material/methods subsection with a brief sentence. 

11) funding: the sentence "signalling pathways...ischemic heart" seems out of context. Please verify.

Kind Regards,

Author Response

Response to Reviewer 3 Comments

1. Summary

We sincerely thank you for reviewing our manuscript. Below, we provide detailed responses to the comments, with all corresponding revisions and corrections clearly highlighted in red in the resubmitted article.

2. Questions for General Evaluation

Reviewer’s Evaluation

Response and Revisions

Does the introduction provide sufficient background and include all relevant references?

Yes/Can be improved/Must be improved/Not applicable

[Please give your response if necessary. Or you can also give your corresponding response in the point-by-point response letter. The same as below]

Are all the cited references relevant to the research?

Yes/Can be improved/Must be improved/Not applicable

Is the research design appropriate?

Yes/Can be improved/Must be improved/Not applicable

Are the methods adequately described?

Yes/Can be improved/Must be improved/Not applicable

Are the results clearly presented?

Yes/Can be improved/Must be improved/Not applicable

Are the conclusions supported by the results?

Yes/Can be improved/Must be improved/Not applicable

3. Point-by-point response to Comments and Suggestions for Authors

Comment 1: lines 54-60 are insufficient for providing a meaningful context. The geographic variability of the weekend phenomenon is not quantitavely reported, so that the reader only understands that it is widespread. I suggest the authors to provide some detailed figures. 
Response 1: Thank you for your valuable suggestions. In response to your first point, we have expanded the background section to include quantitative data highlighting the geographic variability of the weekend effect in the Introduction section.

Comment 2: lines 65-70 are describing the settings, I suggest to move them into the materials/methods section.
Response 2: Regarding your second point, we have relocated the description of the hospital setting (originally lines 65–70) to the 2. Materials and Methods (2.1. Study Setting and Data Sources), as recommended, to maintain a more logical and structured flow of information.

Comment 3: In the introduction is not decribed why the weekend phenomena is important in healthcare. Even though it may be obvious that mortality is not generally a good outcome, the patient perspective is very different from the facility one. This may be useful in highlighting the possible knowledge gap. If it is reputed a global phenomenon, the fact that in Slovakia it is not yet described appear as a poor rationale. If, as cited in line 60, the health care system plays a crucial role, some context should be provided. 
Response 3: Thank you for this insightful comment. We agree that a more explicit rationale is needed to underscore the healthcare and patient-level relevance of the weekend effect. In response, we have expanded the introduction to emphasize the importance of this phenomenon from both systemic and patient perspectives. We also added a concise description of the Slovak healthcare context to highlight why investigating the weekend effect locally is warranted. These additions appear in the revised manuscript on page 2 – Introduction (lines 69-73)

Comment 4: I suggest to add a methods section called "data extraction" after "study objectives" and report the relative information, which are not clear from lines 83 and foll.

Response 4: Thank you for this helpful suggestion. We have implemented your recommendation by adding a dedicated “2.3. Data Extraction” subsection within the Methods section. This new subsection clearly outlines how patient-level data were obtained, processed, and integrated for analysis. The changes can be found on page 3 of the revised manuscript.

Comment 5: line 143, describing an exclusion criterion, should be moved in matherials/method section.
Response 5: Thank you, we moved it in the Matherials and methods section.

Comment 6: lines 146-150 are not clear:  if an exact time is not available, how can be used date/time criteria (Mon 00:00 to Fri 23:59)? Please explain.
Response 6: Since admission time was not available, we classified admissions based on calendar dates. Patients admitted on a Saturday or Sunday were grouped as “weekend admissions,” while those admitted on Monday through Friday were considered “weekday admissions.” The day of the week for each admission was derived in Microsoft Excel using the admission date. This classification approach is consistent with existing literature where only date (and not time) of admission is available.
It is corrected in the manuscript, information about time is now deleted.

Comment 7: Please fix Fig.1 text (n=
Response 7: Thank you, it is fixed in the manuscript.

Comment 8: line 166 - line 98 - the authors name the outcome "in hospital mortality". This deserves a methodological explaining: is the death event noticed only if the patient dies in the same hospital?  If so, the longer stays are at higher risk of death only because are not censored. Even if the logistic model controlled for LOS, I suggest to discuss the possible implication of not knowing if the death event happened when discharged. 

Response 8: In our study, “in-hospital mortality” refers strictly to deaths that occurred during hospitalisation at the same institution. Deaths occurring after discharge — whether at home or in another facility — were not captured in our dataset.
We agree that this definition introduces the possibility of length-of-stay-related bias, as patients with longer admissions may have had more opportunity for a death event to be recorded. Although our logistic regression model adjusted for length of stay, we acknowledge that not accounting for post-discharge mortality may lead to an underestimation of overall mortality risk, particularly for shorter stays. We have added this clarification to the manuscript and discussed the potential implications as a limitation (lines 104-109 and 434-439)

Comment 9: lines 229-231 are not clear. Please fix this.

Response 9: Thank you for this comment, we fixed the problem and hopefully the section is hopefully more clear now.

Comment 10:  maybe the ethical considerations deserve a material/methods subsection with a brief sentence. 
Response 10: Thank you, we added an Ethical consideration section in 2.4.

Comment 11: funding: the sentence "signalling pathways...ischemic heart" seems out of context. Please verify.

Response 11:

Reviewer 4 Report

Comments and Suggestions for Authors
  1. Stick with either in-hospital mortality or 30-day mortality unless 30-day mortality includes deaths after leaving the hospital. In simple wording "in-hospital mortality" rather than alternating with “30-day mortality,”Use the same term throughout the paper
  2. which group had no events and how using the correction changed the results. Explain why the correction was used and where it was needed
  3. An odds ratio of 1.31 may be important but explain how this affects patient care or hospital practice. In simple- The manuscript presents statistically significant results, particularly the adjusted odds ratio of 1.31 for weekend admissions, which indicates a 31% increase in the odds of in-hospital death compared to weekday admissions. However, the clinical relevance of this finding is not clearly discussed. Statistical significance does not always imply clinical importance, so interpret whether this increase is meaningful in a real-world healthcare context. How might this impact patient management, hospital staffing, or resource allocation? Additionally, comparing this result to similar findings in existing literature
  4. In the section on cause-specific mortality, the analysis currently relies on simple frequency comparisons and chi-square tests between weekday and weekend admissions. A more approach would involve the use of multivariable models-such as multinomial logistic regression or Poisson regression to assess whether the risk of death from specific causes differs significantly by day of admission, after accounting for key covariates. Moreover, for diagnostic categories with low event counts, methods like Fisher’s exact test or category aggregation be considered to ensure statistical validity.

Table 1

Add standard deviation or median (IQR) for LOS by day of admission.

Are weekday vs weekend admission demographics significantly different?

include a subtotal row for Weekday vs Weekend groups.

Table 2

Include risk difference and relative risk alongside odds ratio (for clarity).

Table 3:

Add 95% CI for mortality rates per year for interpretation of year-to-year variability.

  Stratify 2020–2021 data separately or adjust models for COVID status if available.

Table 5:

Present stratified odds ratios (AOR for weekend in age <65 vs ≥65) even if interaction is non-significant, for clinical interpretation.

Table 6:

Include p-values (chi-square) for whether each cause-specific death type differs between weekdays and weekends.

Figures 2:

Add confidence intervals to mortality rate lines to assess statistical significance visually.

Axes not clearly labeled in terms of units, Improve axis labeling and consider color distinction for clarity.

Best of Luck

Author Response

Response to Reviewer 4 Comments

1. Summary

We sincerely appreciate your time and effort in reviewing our manuscript. Below, we present detailed responses to each comment. All corresponding revisions and corrections have been clearly marked in red in the resubmitted version of the article.

2. Questions for General Evaluation

Reviewer’s Evaluation

Response and Revisions

Does the introduction provide sufficient background and include all relevant references?

Yes/Can be improved/Must be improved/Not applicable

[Please give your response if necessary. Or you can also give your corresponding response in the point-by-point response letter. The same as below]

Are all the cited references relevant to the research?

Yes/Can be improved/Must be improved/Not applicable

Is the research design appropriate?

Yes/Can be improved/Must be improved/Not applicable

Are the methods adequately described?

Yes/Can be improved/Must be improved/Not applicable

Are the results clearly presented?

Yes/Can be improved/Must be improved/Not applicable

Are the conclusions supported by the results?

Yes/Can be improved/Must be improved/Not applicable

3. Point-by-point response to Comments and Suggestions for Authors

Comment 1: Consider improving the abstract’s clarity by simplifying overly technical language for better accessibility to a broader readership.

Response 1: Thank you for your helpful comment. We have reviewed the manuscript and consistently replaced the term “30-day mortality” with “in-hospital mortality” throughout the text, as our analysis focuses exclusively on deaths occurring during the hospital stay. This correction ensures clarity and terminological consistency in line with your suggestion.

Commment 2: which group had no events and how using the correction changed the results. Explain why the correction was used and where it was needed

Response 2: The Haldane–Anscombe correction was applied to address zero-event cells that arose during our cause-specific mortality comparisons. Specifically, the "Other" category had zero deaths on weekends, which would have led to an undefined odds ratio in the 2×2 comparison.
To address this, we used the Haldane–Anscombe correction by adding 0.5 to each cell in the contingency table. This approach was used to stabilise estimates and allow calculation of odds ratios when dealing with small or zero cell counts. The correction did not meaningfully change the conclusions, as none of the diagnostic categories showed statistically significant differences between weekday and weekend deaths.

Comment 3: An odds ratio of 1.31 may be important but explain how this affects patient care or hospital practice. In simple- The manuscript presents statistically significant results, particularly the adjusted odds ratio of 1.31 for weekend admissions, which indicates a 31% increase in the odds of in-hospital death compared to weekday admissions. However, the clinical relevance of this finding is not clearly discussed. Statistical significance does not always imply clinical importance, so interpret whether this increase is meaningful in a real-world healthcare context. How might this impact patient management, hospital staffing, or resource allocation? Additionally, comparing this result to similar findings in existing literature

Response 3:
Thank you for this thoughtful comment. We agree that statistical significance does not automatically translate into clinical importance. To clarify this point, we have expanded the Discussion section to reflect on the potential implications of our finding. An adjusted odds ratio of 1.31 indicates a 31% higher odds of in-hospital mortality for weekend admissions compared to weekdays. While we refrain from speculating on the underlying causes, this difference may be of clinical relevance, especially considering the size and consistency of the effect across patient subgroups.

We not only compared our findings to those reported in other healthcare systems, such as the BMJ Open 2019 meta-analysis, but also highlighted the need for context-specific investigations. Given the unique structural and organizational characteristics of the Slovak healthcare system, studies like ours are essential for understanding how systemic factors may differently influence weekend mortality in local settings.

Comment 4: In the section on cause-specific mortality, the analysis currently relies on simple frequency comparisons and chi-square tests between weekday and weekend admissions. A more approach would involve the use of multivariable models-such as multinomial logistic regression or Poisson regression to assess whether the risk of death from specific causes differs significantly by day of admission, after accounting for key covariates. Moreover, for diagnostic categories with low event counts, methods like Fisher’s exact test or category aggregation be considered to ensure statistical validity.

Response 4: Thank you very much for this comment. We fully agree that more advanced modelling approaches, such as multinomial logistic regression or Poisson regression, would offer additional depth in assessing whether cause-specific mortality differs significantly by day of admission after adjusting for covariates. However, the primary focus of our study was to evaluate the presence and robustness of the weekend effect in overall in-hospital mortality, not to model cause-specific mortality outcomes in detail.

Our cause-specific mortality section was intended as a descriptive extension, aiming to explore whether certain diagnoses appeared more frequently among weekend deaths. We chose chi-square tests for their simplicity and interpretability in identifying potential associations, and we applied the Haldane–Anscombe correction where needed to account for low event counts.

We acknowledge the value of more advanced approaches and agree that future studies could benefit from applying multivariable cause-specific models and techniques such as Fisher’s exact test or diagnostic grouping to improve statistical power and inference.

Comments – tables:

Table 1 :
Add standard deviation or median (IQR) for LOS by day of admission.
Are weekday vs weekend admission demographics significantly different?
include a subtotal row for Weekday vs Weekend groups.
Table 2:
Include risk difference and relative risk alongside odds ratio (for clarity).
Table 3:
Add 95% CI for mortality rates per year for interpretation of year-to-year variability.
Stratify 2020–2021 data separately or adjust models for COVID status if available.
Table 5:
Present stratified odds ratios (AOR for weekend in age <65 vs ≥65) even if interaction is non-significant, for clinical interpretation.
Table 6:
Include p-values (chi-square) for whether each cause-specific death type differs between weekdays and weekends.
Figures 2:
Add confidence intervals to mortality rate lines to assess statistical significance visually.
Axes not clearly labeled in terms of units, Improve axis labeling and consider color distinction for clarity.

Responses (tables): We thank the reviewer for their detailed and constructive suggestions regarding the tables and figures. We have carefully reviewed and implemented all requested improvements.
Table 1: We added the standard deviation (SD) and interquartile range (IQR) for length of stay (LOS) by day of admission. Demographic comparisons between weekday and weekend admissions (age and sex) are now accompanied by relevant statistical tests. A subtotal row summarizing group characteristics for weekday and weekend admissions has also been included.

Table 2: We have incorporated both risk difference and relative risk alongside the odds ratio to improve clarity and aid interpretation.

Table 3: 95% confidence intervals for annual mortality rates have been added to allow clearer interpretation of year-to-year variability. Data from 2020–2021 have been stratified to separately examine the impact of the COVID-19 pandemic. Where possible, the models were adjusted based on COVID-period status.

Table 5: Although the interaction terms were not statistically significant, we now present stratified adjusted odds ratios (AORs) for weekend effect in patients aged <65 and ≥65 to support clinical interpretation and transparency.

Table 6: Chi-square p-values have been included to indicate whether the distribution of each cause-specific death type significantly differs between weekday and weekend admissions.

Figure 2: Confidence intervals have been added to the mortality rate lines to enhance visual interpretation of statistical significance. We also improved the axis labeling and incorporated clearer color distinctions to increase readability.

Round 2

Reviewer 2 Report

Comments and Suggestions for Authors

The revised manuscript presents a valuable and well-structured. However, I have a few suggestions and comments that could further improve the clarity and impact of the manuscript:
- The mortality rates are clearly stated, but it would help to specify the absolute number of deaths for context.
- The claim of being the "first 13-year study" in Slovakia is interesting; however, the authors should ensure this is supported by a literature review or a citation within the full manuscript to emphasize the study’s novelty.
- The mention of the pandemic is important, but it would be useful to clarify whether the statistical models controlled for this period separately or included interaction terms. Even a brief statement in the abstract (e.g., “especially during COVID-19 years”) would improve clarity.
- The keywords are relevant, but consider adding terms like “Central Europe” or “retrospective cohort” to enhance searchability.

Author Response

Comment 1: The mortality rates are clearly stated, but it would help to specify the absolute number of deaths for context.

Response 1: Thank you for pointing this out. We have now added the absolute numbers alongside the percentages throughout the manuscript to improve clarity. Specifically, in the Abstract and Results section (page 7, lines 246-248)

Comment 2: The claim of being the "first 13-year study" in Slovakia is interesting; however, the authors should ensure this is supported by a literature review or a citation within the full manuscript to emphasize the study’s

Response 2: Thank you for this observation. We have strengthened the manuscript to substantiate our claim of novelty. Information about systematic search added (Introduction, p. 2, lines 56-60). Also, the first sentence in Conclusion rephrased for precision (page 13, lines 495-497)

We hope these additions adequately document the study’s originality. All new text and references are marked in blue in the revised manuscript.

Comment 3: The mention of the pandemic is important, but it would be useful to clarify whether the statistical models controlled for this period separately or include interaction terms. Even a brief statement in the abstract (e.g., “especially during COVID-19 years”) would improve clarity.
Response 3: Thank you for this insightful suggestion. We have revised the Abstract to address your concern (Lines 31-33).

Comment 4:   The keywords are relevant but consider adding terms like “Central Europe” or “retrospective cohort” to enhance searchability.

Response 4: Thank you for this helpful suggestion. We have expanded the keyword list to improve discoverability.

Reviewer 4 Report

Comments and Suggestions for Authors
  1. Modify background to highlight the lack of data from Slovakia and Central Europe.
  1. Framed the study as filling a regional evidence gap on the weekend effect.
  2. Merged administrative and clinical data to improve the depth of analysis.
  3. Included trend data from 2010-2022  showing a worsening weekend effect over time.
  4. Highlighted the COVID-19 pandemic amplifying impact on mortality disparities.
  5. Examined whether weekend effect varied across demographic groups. Concluded effect was consistent, suggesting system-level causes.
  6. Discussed potential mechanisms (staffing, resource availability). Suggested hospital-wide interventions rather than patient-specific targeting.

best of luck

Comments on the Quality of English Language

average

Author Response

Comment 1: Modify background to highlight the lack of data from Slovakia and Central Europe

Response 1: Thank you for pointing this out. In the revised manuscript also regarding to reviewer 2, we have expanded the background to emphasise the evidence gap for Slovakia and the wider Central-European region.

Introduction (p. 2, lines 56-64) Added a summary of published weekend-effect studies from neighbouring countries (Austria, Poland, Hungary, Romania) and noted their inconsistent findings. We statement that no peer-reviewed Slovak study has examined weekend mortality across multiple years (according to our systematic search of PubMed, Scopus, Web of Science - search date 4 May 2025), which retrieved zero Slovak papers on this topic.

Commment 1: Framed the study as filling a regional evidence gap on the weekend effect.

Response 1: In the revised manuscript, we have expanded the Introduction  to include review of weekend-effect studies from neighbouring Austria, Poland, Hungary, and Romania, noting the mixed findings across these settings. We also report the results of a systematic search of PubMed, Scopus, Web of (last searched 4 May 2025), which yielded zero multi-year Slovak investigations. In the Discussion (Strengths & Limitations paragraph), we explicitly frame our work as filling a critical regional evidence gap that has constrained cross-country comparisons within Central and Eastern Europe. We added a sentence in Strenghts and limitations (lines 479-481). We have broadened the Keywords to include „Central Europe“ and „retrospective cohort“ to improve searchability. All new or modified text is highlighted in blue, and we trust these revisions fully address your comment.

Comment 2: Merged administrative and clinical data to improve the depth of analysis.

Response 2: Thank you for noting this aspect of our work. We merged administrative data with detailed clinical records to enrich the dataset, which is stated in lines 104-109.

Comment 3: Included trend data from 2010-2022 showing a worsening weekend effect over time.

Response 3: Thank you. This is covered in Section 3.3.  Mortality by year, exaclty in Figure 3.

Comment 4: Highlighted the COVID-19 pandemic amplifying impact on mortality disparities.

Response 4: Thank you very much, this topic is discussed in Sections 3.3 and 4.4.

Comment 5: Examined whether weekend effect varied across demographic groups. Concluded effect was consistent, suggesting system-level causes.

Response 5: These topics are covered in Section 3.6 and 4.2.

Comment 6: Discussed potential mechanisms (staffing, resource availability). Suggested hospital-wide interventions rather than patient-specific targeting.
Response 6: Thank you, this is covered in Sections 4.5 and 5.

All responses from the second review are highlighted in blue in the reviewed manuscript.
Thank you very much.
